# Association of Postural Instability with Autonomic Dysfunction in Early Parkinson’s Disease

**DOI:** 10.3390/jcm9113786

**Published:** 2020-11-23

**Authors:** Sooyeoun You, Hyun Ah Kim, Hyung Lee

**Affiliations:** Department of Neurology, Keimyung University School of Medicine, 1035, Dalgubeol-daero, Dalseo-gu, Daegu 42601, Korea; omoi81@hanmail.net (S.Y.); kha0206@dsmc.or.kr (H.A.K.)

**Keywords:** Parkinson’s disease, posturography, autonomic function test, heart rate variability, correlation, parasympathetic activity

## Abstract

Background: There have been several pathologic data that support an association between postural instability (PI) and autonomic dysfunction in Parkinson’s disease (PD). The purpose of this study was to investigate the correlation of PI and autonomic deficits in early PD. Methods: We collected 17 patients with a diagnosis of early PD. PI was assessed by computerized dynamic posturography (CDP). Standardized autonomic function test (AFT) and time and frequency domain spectral analysis of heart rate variability (HRV) were performed. CDP data obtained from the 21 patients were compared to that from age- and sex-matched healthy controls. We collected HRV data from 18 other age- and sex-matched controls. All patients were evaluated in the “OFF” state. We used Mann–Whitney U-test to compare parameters of CDP between the early PD and control groups. Spearman correlation was used for correlation analysis between parameters of CDP and autonomic function test in PD patients. Results: Most patients (76.5%) showed mild or moderate autonomic dysfunction in the standardized AFT. In CDP, sensory ratios of equilibrium score (e.g., visual and vestibular) and composite scores were significantly lower in PD patients than in controls. In HRV, the low-frequency/high-frequency ratio during the tilt and the gap of low- frequency/high-frequency ratio from supine to tilt were significantly different in both groups. The parameters of time and frequency domains of HRV reflecting parasympathetic function were correlated with equilibrium scores for somatosensory organization test in CDP. Discussion: PI was associated with parasympathetic autonomic dysfunction in early PD. This result was in accordance with a previous assumption that PI in PD is related to parasympathetic cholinergic neuron loss in the brainstem.

## 1. Introduction

Postural instability is one of the cardinal motor symptoms in Parkinson’s disease (PD), especially in the late stage. However, according to previous studies [1,2,3,4,5,6,7], patients with early PD who showed a normal result at the pull test in the bedside examination revealed a mild postural instability in both static and dynamic posturography. Autonomic dysfunction is a well-known non-motor symptom of PD. Adrenergic autonomic dysfunction, such as orthostatic hypotension (OH), increases the risk of falling in PD [8]. In PD patients with neurogenic OH, the number of fall-related emergency room visits or hospitalizations is higher than in PD patients without OH [8]. Postural instability and autonomic dysfunction are thought to be caused by damage in non-dopaminergic systems. They are known to be associated with the severity of disease across the disease course of PD [9]. Known brainstem pathologic data may give a clue to explain an association between the postural instability and autonomic dysfunction in PD [10,11,12,13]. In previous studies in PD [14,15,16], inclusion pathologies in the brainstem nuclei such as Lewy body and Lewy neuritis were prevalent in the brainstem such as in the substantia nigra; ventral tegmental area; pedunculopontine and raphe nuclei; periaqueductal gray; locus ceruleus; parabrachial nuclei; reticular formation; prepositus hypoglossal, dorsal motor vagal and solitary nuclei, most of which are known to participate in locomotor postural control and autonomic regulation [12,16]. Therefore, we can reasonably assume an association between postural instability, a less well-explained motor symptom in PD, and autonomic dysfunction. Thus, we performed this study to investigate the correlation of postural instability assessed with computerized dynamic posturography (CDP) and autonomic deficits assessed with various autonomic function tests in patients with early PD. In PD patients, observation in the OFF state aimed to measure autonomic function in a condition that excludes levodopa’s effects as much as possible. Since the OFF state can be a burden to PD patients, we included PD patients of H and Y stage 1 and 2 who do not have severe motor symptoms even in the OFF state. Because the degree of autonomic dysfunction in early PD estimated with a battery of standardized autonomic function test is known to be mild [17], we added time and frequency domains spectral analysis of heart rate variability (HRV), a powerful and reliable tool for assessing the fluctuations in the autonomic nervous system and balance of sympathetic and parasympathetic activities [18,19].

## 2. Experimental Section

### 2.1. Patients

The present study was a retrospective and medical chart review research. From September 2013 to July 2018, we examined the medical records of patients with early PD.

The diagnosis of PD was assessed clinically by the U.K. Brain Bank criteria. The early stage of PD was defined with the modified Hoehn and Yahr (H and Y) Stage 1 or 2 [20]. The severity of motor symptoms of PD was assessed by Movement Disorder Society Unified Parkinson’s Disease Rating Scale motor part 3 (MDS-UPDRS-III) [21]. All patients had taken dopaminergic medications for symptomatic treatment of PD. We only included early PD patients with no other concurrent medical illnesses or conditions that potentially cause autonomic dysfunction.

Patients with cognitive deficits (Mini-Mental State Examination <23), severe dyskinesia (Goetz score >3), unpredictable motor fluctuation, or an orthopedic problem were excluded. We also excluded patients who had current dizziness/vertigo or had a history of peripheral or central vestibulopathy. Finally, 17 patients with early PD and on a stable dosage of dopamine drugs were studied. CDP data from 21 age- and sex-matched healthy controls were collected from our normal data pool for vestibular study at Keimyung University Vestibular Research center. We also got the data of 18 age- and sex-matched healthy volunteers from our normal data pool for autonomic study at Keimyung University Autonomic Laboratory center as controls for HRV. We excluded early PD patients with medical illnesses or conditions such as multisystem atrophy, diabetic neuropathy, hypovolemia (dehydration), anemia, or heart failure that potentially cause autonomic dysfunction. Furthermore, we excluded patients with early PD who were taking medication such as antidepressant with serotonin reuptake inhibitors for benign prostatic hypertrophy because these drugs also may affect autonomic function. Neurological examination and other investigations for PD patients were performed in the “OFF” state.

### 2.2. CDP

We used CDP (EquiTest version 4.0, NeuroCom, Clackamas, OR, USA) to assess postural instability quantitatively. It consists of a footplate with a visual surround and a software program, which measures and records the vertical and horizontal forces exerted by the participant’s feet. The assessment has two parts: the sensory organization test (SOT) and the motor control test (MCT). The participants were asked to put their bare feet parallel on the footplate while staring straight ahead during testing. CDP was conducted by two experienced neuro-otologists (H.L., H.A.K.), who were blinded to the information of participants.

#### 2.2.1. Somatosensory Organization Test (SOT)

The SOT assesses the ability of participants to process individual sensory input to maintain balance control under the combined conditions of eye-opening status, the sway of footplate, or movement of visual surrounds. Participants were tested under six sensory conditions (three 20 s trials each), where surface and visual surroundings were systematically modified: (1) eyes-open with fixed footplate and visual surround, (2) eyes-closed with fixed footplate and visual surround, (3) eyes-open with fixed footplate and moving visual surround, (4) eyes-open with sway of the footplate and fixed visual surround, (5) eyes-closed with sway of the footplate and fixed visual surround, (6) eyes-open with sway of the footplate and visual surround (Figure 1). An equilibrium composite score was calculated by comparing the angular difference between a participant’s maximum anterior to the posterior center of gravity displacement and the maximum possible sway range of 12.5°. A score of 0 indicates a fall, whereas a score of 100 means no sway. A strategy score presents the degree of use of ankle strategy compared to the hip strategy for maintaining balance.

#### 2.2.2. Motor Organization Test (MCT)

The purpose of the MCT is to evaluate the ability of the motor control system to recover from an unpredictable forward or backward perturbation. The MCT measures the symmetry of both legs’ power, the latency of motor response, the strength of the active force, and the adaptation to repetitive situations. When the footplate moves forward or backward with different speeds of 5 cm/s (small), 10 cm/s (medium), or 15 cm/s (large), the time in milliseconds from the movement of the footplate to initiation of the active force response in a leg was measured, which is defined as latency.

### 2.3. Standardized Autonomic Function Tests

The quantitative sudomotor axon reflex test, tilt table test, Valsalva maneuver (VM), and HR response to deep breathing were performed to evaluate sympathetic and parasympathetic autonomic function. Beat-to-beat blood pressure (BP) and heart rate responses were measured noninvasively using the Finometer device (Finapres Medical Systems BV, Amsterdam, The Netherlands). The tilt protocol included 10 min in the supine position and 20 min of tilt at 70 degrees. Orthostatic hypotension (OH) was defined as a decrease in systolic BP of at least 20 mmHg or a decrease in diastolic BP of at least ten mmHg between supine rest for 10 min and an upright posture for 20 min. In addition to BP monitoring using a Finometer, BP was obtained at baseline and at every minute during the tilt using a manual sphygmomanometer (Tycos, Skaneateles Falls, NY, USA). Detailed testing techniques have been addressed in our previous reports [22,23]. The composite autonomic severity score (CASS) was derived from the autonomic reflex screen, as described previously [24].

### 2.4. Heart Rate Variability (HRV) Test

On a standard electrocardiogram (ECG) (SEER Light compact digital Holter recorder; GE Healthcare), the maximum upward deflection of a normal QRS complex is at the peak of the R-wave, and the duration between two adjacent R-wave peaks is termed as the R–R interval. ECG signals were visually inspected for maximum ectopic beats and processed using commercial software. The time domain analysis comprised of different measures of the normal-to-normal (NN) intervals, including standard deviation of the NN intervals (SDNN), square root of the mean squared difference of successive NN intervals (rMSSD), number of pairs of adjacent NN intervals differing by more than 50 ms in the entire recording divided by the total number of all NN intervals (pNN50), and total number of all NN intervals divided by the height of the histogram of all NN intervals measured on a discrete scale with bins of 7·8125 ms (1/128 s) (HRV triangular index). Frequency domain analysis was estimated by how the power (variance) distributes as a function of frequency and its fluctuation. Power spectral density was analyzed over 10 min at low frequency (LF, 0.04–0.15 Hz) and at high frequency (HF, 0.15–0.40 Hz) according to recommendations of Task Force of the European Society of Cardiology and the North American Society of Pacing and Electrophysiology [20,21]. Normalized LF, Normalized HF, and LF/HF ratio were also calculated. Various time and frequency domains of HRV were recorded under the following conditions: (1) resting supine during spontaneous breathing (10–15 breathing per minute) for 10 min and (2) 10 min passive tilting at 70 degrees.

All autonomic function tests, including HRV, were performed under standardized conditions; the dopaminergic drugs were stopped two days before testing, and all subjects were investigated after fasting for at least 4 h. The autonomic function test was performed under constant environmental conditions (i.e., quiet, an ambient temperature of 73–76 F, and humidity kept at approximately 50%). No coffee, food, or nicotine was permitted for six hours before the study. All patients performed autonomic function tests and CDP on the same day.

### 2.5. Statistical Analysis

We used SPSS (Statistical Package from Social Science Program) version 21.0 for data processing (SPSS Inc., Chicago, IL, USA). Quantitative data were presented as mean ± standard deviation (SD). We used Mann–Whitney U-test to compare parameters of CDP between the early PD and control groups. Spearman correlation was used for correlation analysis between parameters of CDP and autonomic function test in PD patients. All tests were two-tailed and considered statistically significant at *p* < 0.05. The Institutional Review Board of Keimyung University Dongsan Medical Center approved this study.

## 3. Results

### 3.1. Patients and Controls

Of the 17 patients, 10 (59%) were women. The mean age was 67.7 ± 8.4 years, and the mean disease duration was 3.6 ± 3.0 years (9 months–11 years). The mean levodopa equivalent dose was 412 ± 232 mg. The score of UPDRS part III at OFF time was 28.4 ± 11.8. All patients were H and Y stage 2. Among 17 patients with early PD, all of the early PD was tremor-dominant (*n* = 15) or akinetic rigidity type (*n* = 2). None of the patients showed postural instability and gait disturbance in pull test and tandem gait test. The mean age of 21 healthy controls for CDP was 64.3 ± 5.2 years. Twelve of 21 (57%) were women. The mean age of 18 healthy controls for HRV was 62.1 ± 10.5 years. Nine of 18 (50%) were women.

### 3.2. CDP

In PD, the strategy score of SOT and latency of MCT were within normal ranges. However, when sensory ratios of equilibrium score (e.g., visual and vestibular) and a composite score for SOT were compared, the scores were significantly lower in PD patients than in healthy controls (Table 1).

### 3.3. Standardized Autonomic Function Tests

Approximately 60% (10/17) of patients showed autonomic dysfunction in standardized autonomic function tests. Eight patients had mild (3 or less on CASS) autonomic dysfunction. Others showed moderate autonomic dysfunction (4 to 6 points on CASS). Deficits were predominantly adrenergic (5/17, 29%) and sudomotor (5/17, 29%) dysfunctions. Seven of 17 patients (41%) showed OH during the tilt. In VM, five patients showed decreased or absented late phase II, and four had abnormal phase IV overshooting. Five patients showed prolonged BP recovery time. Cardiovagal dysfunction was observed in three (3/17, 18%) patients. Three patients showed an abnormal Valsalva ratio, and one had an abnormal HR response to deep breathing. There were no significant correlations between parameters for SOT or MCT in CDP and parasympathetic cardiovagal, sympathetic adrenergic, or sympathetic sudomotor-related parameters from standardized autonomic function tests. By definition, none of the controls showed abnormality.

### 3.4. HRV

The time and frequency domain spectral analysis results of the HRV test in early PD patients and controls are shown in Table 2. There were no significant differences in the HRV test results between PD patients and healthy controls except for the LF/HF ratio. However, rMSSD, pNN50, HRV triangular, and HF norm tended to be decreased during the tilt compared to that in the supine position in controls, but those were not reduced in the PD group. LF norm and LF/HF were increased during the tilt compared to those in the supine position in controls, but those were not increased in early PD patients. LF/HF ratio during the tilt was significantly lower in PD patients than in controls (*p* = 0.021). The gap of LF/HF from supine to tilt was a negative value in controls but was positive in PD groups and was significantly different in both groups (*p* = 0.015).

### 3.5. Correlations between Equilibrium Scores for CDP in CDP and HRV Parameters

pNN50 during supine position showed a positive correlation with vestibular and composite SOT scores. HF during supine also showed a positive correlation with the composite SOT score. rMSSD during the tilt showed a negative correlation with the visual preference SOT score. pNN50 during the tilt also showed a negative correlation with somatosensory SOT and visual preference SOT scores. There was a positive correlation between the gap of RMSSD, pNN50, triangular HRV and HF from supine to tilt, and composite SOT score in early PD patients (Figure 2). pNN50 also showed a positive correlation with vestibular SOT (Table 3).

## 4. Discussion

This is the first study to investigate the correlation between postural instability and autonomic dysfunctions in patients with early PD. The main finding in our study was that there is a significant correlation between parasympathetic autonomic abnormalities shown as rMSSD, pNN50, HF, and various equilibrium scores for SOT in CDP. This result is in accordance with previous studies that postural instability in PD is associated with brainstem parasympathetic cholinergic neuronal loss [25,26].

Our key finding on a significant correlation between the sensory ratios of equilibrium score such as somatosensory, vestibular, visual preference, and composite score for SOT, and parasympathetic autonomic abnormality as estimated with HRV also supports the hypothesis that the parasympathetic cholinergic system in the brainstem such as dorsal glossopharyngeus–vagus complex and pedunculopontine nucleus (PPN) may be responsible for postural control in PD [27,28]. Although postural instability or falling is known to be a manifestation of later or advanced PD, several studies reported on postural instability or falling in early stage PD [29,30,31,32]. Previous studies have shown that age, postural instability, gait disturbance subtype, visual deprivation, and cognitive function are related to postural instability in PD [1,8,9,31,32]. Similar to previous studies [2,3,4,7], we found that even in early PD without gait problem, mild postural instability was found in dynamic posturography compared to controls. However, previous studies on postural instability as assessed with dynamic posturography in early PD demonstrated contradictory findings. Some studies revealed a significant correlation between parameters for any MCT in dynamic posturography and postural instability in early PD [1,5]. Others showed that parameters for any SOT in dynamic posturography and postural instability in early PD were significantly correlated [4]. We found that MCT did not reveal any differences between groups. Otherwise, sensory ratios of equilibrium score (e.g., visual and vestibular) and a composite score for SOT were significantly lower in early PD than in healthy controls, suggesting that early stage PD patients had a relatively preserved efferent motor copy system that is widely recognized as a perfect reproduction of motor commands, and the resultant prediction is widely presumed to represent the precise sensory consequences of each motor act [33]. Postural functions depend in part on the integration of sensory information from visual, proprioceptive, and vestibular systems [34]. Our result is in agreement with previous studies that the integration of relevant sensory information may be affected in patients with PD [35,36].

Autonomic dysfunction in our study was mostly mild, which is in agreement with a previous report that PD patients usually have mild cardiovascular autonomic deficits; even autonomic dysfunction is a well-known non-motor symptom of PD [17]. Our patients had an OH of 41%, which is similar to that (20–50%) reported for idiopathic PD [37,38].

The time and frequency domain spectral analysis of HRV testing is a well-documented tool for evaluating cardiac autonomic function, which represents a balanced cardiac control mechanism by recurrent changes in R–R interval characteristics [18,19]. Because the efferent vagal activity is a significant contributor to the HF component, the HF component is generally considered as a marker reflecting parasympathetic activity [39]. In contrast, the LF component of the frequency domain of HRV is usually considered as a marker of sympathetic modulation [40] or as a parameter reflecting both sympathetic and vagal activities [41]. Consequently, the LF/HF ratio is considered a sympathovagal balance [18]. The time and frequency domain variables are known to be strongly correlated with each other [18]. For example, RMMSD and pNN50 in time domain variables approximately relate to HF in the frequency domain [18].

The clinical use of HRV in diverse clinical populations, including sudden cardiac death associated with syncope, diabetic neuropathy, sleep apnea, hypertension, and myocardial infarction have been studied [18]. However, a systemic quantification of autonomic deficit assessed with HRV in PD has not been performed. In the present study, the LF/HF ratio during supine position was not different in both groups. However, the LF/HF ratio during the tilt was significantly lower in PD than in controls, suggesting an impairment of sympathovagal balance during upright posture in PD. The rMSSD, pNN50, and HF, which are well-known markers reflecting parasympathetic activity, were significantly correlated with the sensory ratios of equilibrium score such as somatosensory, vestibular, visual preference, and a composite score for SOT. It suggests that impairment in the integration of multimodal sensory (from proprioceptive, vestibular, and visual) information for a normal postural function is related to parasympathetic autonomic dysfunction.

The pathophysiology of postural instability in PD is believed to be mainly related to the non-dopaminergic system, but it is not still clear. The current data have suggested that the integrity of the pedunculopontine nucleus (PPN) cholinergic neurons and thalamic efferents are a keystone for postural sensory integration function in PD, possibly by participating in the integration of multimodal sensory (i.e., visual, vestibular, and somatosensory) input information [42]. It is well known that neurons in the PPN exhibit a wide heterogeneity in terms of their neurochemical nature and their connectivity. The cholinergic neurons are known to be located in the caudal PPN, whereas GABAergic neurons are highly concentrated in the rostral PPN. The caudal cholinergic PPN is also heavily connected to the thalamus [43]. Thus, damaged integrity of cholinergic PPN in the caudal portion and their thalamic efferents is believed to be implicated in postural instability in PD [42]. Our key finding on a significant correlation between the sensory ratios of equilibrium score such as somatosensory, vestibular, visual preference, and a composite score for SOT, and parasympathetic autonomic abnormality as estimated with HRV also supports the hypothesis that the parasympathetic cholinergic system in the brainstem such as the caudal PPN may responsible for postural control in PD.

Our results have some clinical implications. Because postural instability, as assessed with SOT in CDP, was associated with parasympathetic autonomic dysfunction in early PD patients who had a normal result in the pull test in the bedside examination, all patients with PD should be evaluated with the autonomic function test, irrespective of the stage of disease severity. Clinicians should be aware of the possibility of autonomic-dysfunction-related postural instability, even when typical postural instability is absent on routine neurological examination.

Our study had several limitations. First, the sample size was small because we needed to include only early PD patients with no accompanying autonomic, cognitive, orthopedic, or vestibular problems. Further studies with a large number of patients will be required to confirm our findings concerning autonomic dysfunction in early PD. Second, inherent bias is inevitable in the retrospective study design despite the use of consecutively collected registry data. Third, longitudinal studies, including follow-up observations, will be necessary to identify whether CDP and autonomic results deteriorate as the motor symptoms worsen. Fourth, the current definition of early PD is confined to the H and Y stage, but even patients with early PD may experience falling when the disease duration is prolonged [44]. Further studies are needed to determine how the duration of disease (rather than the H and Y stage) affects the postural instability in PD. Fifth, we tried to exclude patients with dementia in early PD because we needed to exclude Lewy body dementia or progressive supranuclear palsy. However, we could not exclude patients with mild cognitive impairment in our study. Finally, we could not perform questionnaires for evaluating autonomic symptoms for early PD patients. In the future study, we could investigate the relationship between autonomic symptoms identified with questionnaires and postural instability in a larger number of patients.

In conclusion, postural instability shown as various equilibrium scores for SOT in CDP seems to be associated with parasympathetic autonomic dysfunction, as assessed with time and frequency domain spectral analysis of HRV in early PD. This result was in accordance with a previous assumption that postural dysfunction in PD is related to the impaired integrity of cholinergic neurons in the caudal PPN in the brainstem.

## Figures and Tables

**Figure 1 jcm-09-03786-f001:**
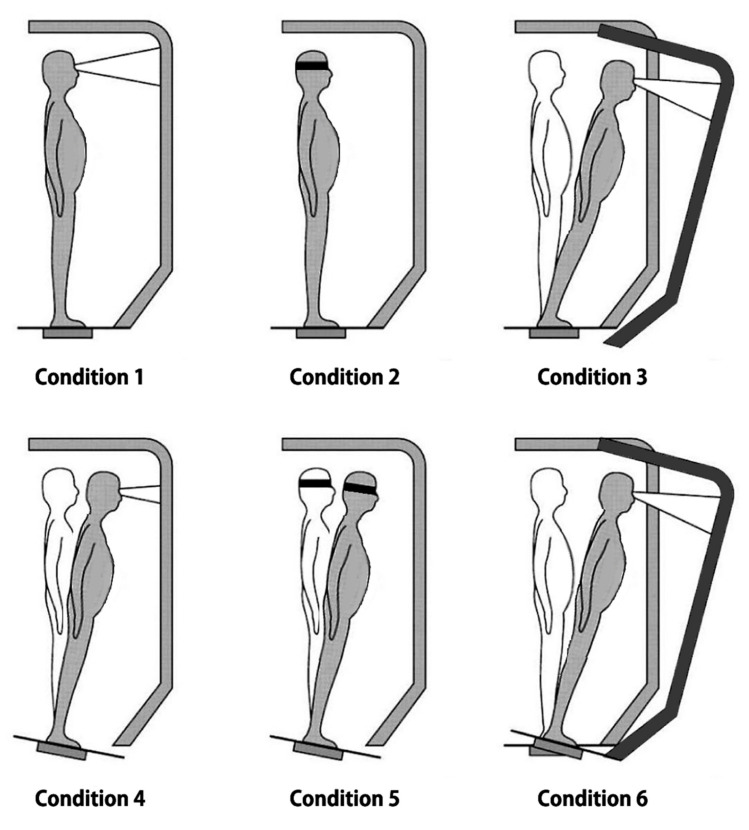
Test conditions for sensory organization test (SOT) 1 to 6 under which equilibrium score was calculated. Under condition 1 (eyes open), and condition 2 (eyes closed), both the platform and the surround remain immobilized. Under condition 3, the surround moves. Under condition 4, the platform moves and the surround remains fixed. Under condition 5, the platform moves while the subject keeps his/her eyes closed. Under condition 6, both the surround and the platform move.

**Figure 2 jcm-09-03786-f002:**
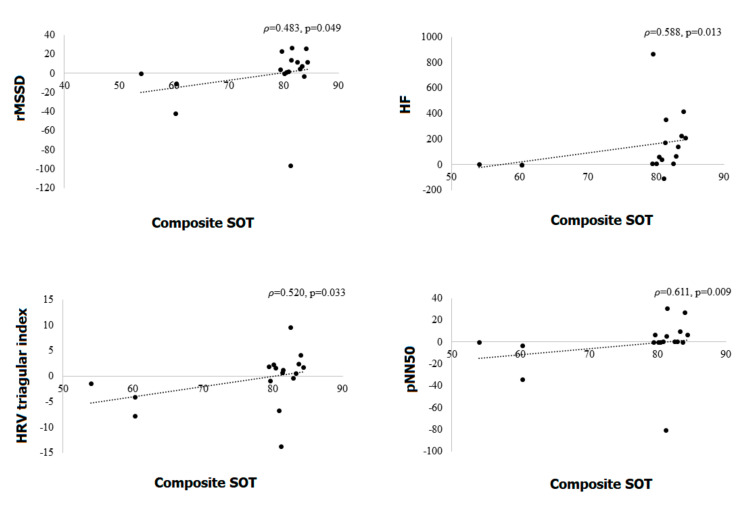
Correlations between the gap of rMSSD, pNN50, triangular HRV and HF from supine to tilt, and composite sensory organization test (SOT) score in early Parkinson’s disease patients. HRV, heart rate variability; rMSSD, the square root of the mean squared difference of successive NNs; pNN50, proportion of NN50 divided by the total number of NN (R–R) intervals; HF, high frequency.

**Table 1 jcm-09-03786-t001:** Comparison of computerized dynamic posturography (CDP) results between early Parkinson’s disease (PD) patients and normal controls.

CDP		PD, *n* = 17 Mean (SD)	95% CI	Normal, *n* = 21 Mean (SD)	95% CI	*U*, *p* Value
Demographics	Age (Years) Sex (Female)	67.7 (8.4) 10 (59%)	63.71–71.7	64.3 (5.2) 12 (57%)	62.08–66.53	0.126 0.800
SOT Equilibrium score	Somatosensory	0.98 (0.02)	0.98–0.99	0.98 (0.03)	0.97–1	169.5, 0.781
	Visual	0.85 (0.09)	0.81–0.9	0.93 (0.07)	0.91–0.96	148.0, **0.004 ^*^**
	Vestibular	0.59 (0.29)	0.46–0.73	0.80 (0.08)	0.77–0.84	151.5, **0.009^ *^**
	Visual preference	0.99 (0.05)	0.97–1.02	1.02 (0.07)	1–1.05	183.0, 0.107
	Composite	77.6 (9.5)	73.09–82.12	80.5 (2.7)	79.35–81.66	357.0, **0.008 ^*^**
SOT Strategy score (mean)	Condition 1	97.7 (2.2)	96.66–98.75	98.1 (1.4)	97.51–98.7	202.0, 0.461
	Condition 2	96.9 (2.8)	95.57–98.24	97.1 (2.7)	95.95–98.26	192.5, 0.797
	Condition 3	96.2 (4.8)	93.92–98.49	97.4 (2.7)	96.25–98.56	214.0, 0.315
	Condition 4	83.6 (5.6)	80.94–86.27	84.0 (5.2)	81.78–86.23	185.0, 0.819
	Condition 5	77.2 (10.9)	72.02–82.39	72.5 (9.7)	68.36–76.65	162.0, 0.164
	Condition 6	79.1 (8.8)	74.92–83.29	75.2 (10.0)	70.93–79.48	151.0, 0.213
	Composite score	88.5 (9.5)	83.99–93.02	87.4 (11.8)	82.36–92.45	160.5, 0.178
MCT Latency, backward	Small, L	141.2 (23.1)	130.22–152.19	151.0 (17.3)	143.61–158.4	143.5, 0.147
	Small, R	160.0 (30.2)	145.65–174.36	151.4 (23.7)	141.27–161.54	147.5, 0.333
	Medium, L	143.1 (34.2)	126.85–159.36	136.2 (12.8)	130.73–141.68	158.5, 0.396
	Medium, R	150.6 (29.5)	136.58–164.63	140.0 (14.8)	133.67–146.34	162.5, 0.159
	Large, L	139.1 (33.7)	123.09–155.12	138.1 (16.3)	131.13–145.08	183.0, 0.909
	Large, R	144.1 (29.6)	130.03–158.18	137.1 (18.5)	129.19–145.02	145.0, 0.380

**^*^***p* < 0.05. SD, standard deviation; CI, confidence interval; DP, computerized dynamic posturography; PD, Parkinson’s disease; SOT, sensory organization test; MCT, motor control test; L, left; R, right.

**Table 2 jcm-09-03786-t002:** Results of time and frequency domain spectral analysis of heart rate variability in early PD patients.

	Mean ± SD	PD (*n* = 17)	95% CI	Controls (*n* = 18)	95% CI	*U*, *p* Value
Demographics	Age (years) Sex (F)	67.7 (8.4) 10 (59%)	63.71–71.7	62.1 (10.5) 9 (50%)	57.25–66.96	0.096 0.860
Supine−tilt	SDNN	−2.0 ± 20.5	−11.75–7.75	−1.4 ± 6.6	−4.45–1.65	130.0, 0.908
	rMSSD	−1.1 ± 29.2	−14.99–12.79	4.2 ± 5.3	1.76–6.65	146.5, 0.477
	pNN50	−1.8 ± 24.4	−13.4–9.8	1.8 ± 2.9	0.47–3.14	151.0, 0.559
	HRV, triangular	−0.5 ± 5.3	−3.02–2.02	0.3 ± 1.8	−0.54–1.14	147.0, 0.554
	LF	183.4 ± 241.9	68.41–298.4	95.5 ± 88.9	54.44–136.57	191.0, 0.173
	LF norm	3.1 ± 32.4	−12.31–18.51	−10.2 ± 17.2	−18.15–2.26	101.0, 0.135
	HF	146.7 ± 231.7	36.56–256.85	96.6 ± 89.0	55.49–137.72	165.0, 0.413
	HF norm	2.8 ± 26.0	−9.56–15.16	10.2 ± 17.2	2.26–18.15	160.0, 0.490
	LF/HF	0.2 ± 2.0	−0.76–1.16	−1.4 ± 1.8	−2.24–−0.57	76.0, **0.015 ^*^**

**^*^***p* < 0.05. SD, standard deviation; CI, confidence interval; PD, Parkinson’s disease; SDNN, standard deviation of the NN intervals; rMSSD, the square root of the mean squared difference of successive NNs; pNN50, proportion of NN50 divided by the total number of NN (R–R) intervals; HRV, heart rate variability; LF, low frequency; HF, high frequency; LF norm, LF power in normalized units; HF norm, HF power in normalized units.

**Table 3 jcm-09-03786-t003:** Correlation analysis for SOT equilibrium scores in CDP and HRV parameters in early PD patients.

		SOM	VIS	VEST	PREP	COM
Supine−tilt	SDNN	0.042	0.038	0.232	0.163	0.442
	rMSSD	0.35	−0.204	0.304	0.015	**0.483 ^*^**
	pNN50	0.244	−0.164	**0.514 ^*^**	−0.152	**0.611 ^**^**
	HRV, triangular	0.47	0.137	0.157	0.35	**0.520 ^*^**
	LF	0.103	0.181	0.214	0.066	0.404
	LF norm	−0.381	0.324	−0.184	0.194	−0.066
	HF	0.319	−0.105	0.482	−0.054	**0.588 ***
	HF norm	0.338	−0.235	0.174	−0.083	0.169
	LF/HF	−0.37	0.321	−0.246	0.194	−0.091

**^*^***p* < 0.05, ******
*p* < 0.01. SOT, sensory organization test; CDP, computerized dynamic posturography; HRV, heart rate variability; PD, Parkinson’s disease; SDNN, standard deviation of the NN intervals; rMSSD, the square root of the mean squared difference of successive NNs; pNN50, he proportion of NN50 divided by the total number of NN (R–R) intervals; LF, low frequency; HF, high frequency; LF norm, LF power in normalized units; HF norm, HF power in normalized units; SOM, somatosensory; VIS, visual; VEST, vestibular; PREP, visual preference; COM, composite.

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
