# Peer review of "Association of Postural Instability with Autonomic Dysfunction in Early Parkinson’s Disease"

_jcm, 2020, doi:10.3390/jcm9113786_

Round 1

Reviewer 1 Report

Overall this research is very interesting and was conducted in a well-organized manner. The paper is well-written; a good read. However, there are major weak spots that either need a more elaborate clarification or a better justification of choices.

In this paper the authors present a study in which they investigated the association of postural instability in early PD with with parasympathetic autonomic dysfunction. 

Perhaps the most useful recommendation I can make that would make the story extremely clear from the outset, early on in the article, is to provide the framework explaining association between the postural instability and autonomic dysfunction in PD.

Line 83, was any observations on using OFF state? since this could lead to emerge symptoms on patients! Or authors should provide justification why they did not use ON state in stead. 

Line 119, 10 min --> 10 minutes 

Line 129, EGC ---> ECG

Line 155, Re-view--> review 

Table 1, I guess that n=21 for the controls should be 19?  

Also the significant values should be highlighted with bold. 

Line 140, Normalized LF/HF ratio were calculated but never explained the min or max scale values what does it mean!

Line 134, I have concerns about the NN intervals, the authors decided to collect data after 10ms and 20ms. Here I can see that intervals from 0-10ms were omitted and could reveal that patients performed worse/better before it gets to 10ms? could you please elaborate on that?

Line 217-224, should be removed! 

The discussion has to be improved and supported with references; 

Line 248, efferent motor copies need to be elaborated!

Line 249, need reference! 

Line 302, earl---> early

I would suggests the authors to have a conclusion section to summarize their results in order to highlight the main findings in the study! 

It would be nice to have figures illustrating the conditions to make it clear for the readers especially with CDP!

Author Response

  1. Perhaps the most useful recommendation I can make that would make the story extremely clear from the outset, early on in the article, is to provide the framework explaining association between the postural instability and autonomic dysfunction in PD.

→ According to the reviewer’s recommendation, we edited the Introduction part for a more precise explanation.

  1. Line 83, was any observations on using OFF state? since this could lead to emerge symptoms on patients! Or authors should provide justification why they did not use ON state instead. 

→ Since the OFF state can be a burden to PD patients, we included PD patients of  H&Y stage 1&2 who do not have severe motor symptoms even in the OFF state. In PD patients, observation on the OFF state aimed to measure autonomic function in a condition that excludes levodopa's effects as much as possible. Previous studies have also reported that taking levodopa immediately (30~60 min) affects the vagal tone. (Annu Int Conf IEEE Eng Med Biol Soc. 2015) (Neurology India, 2011). We added sentences to the Introduction section to explain it.

  1. Line 119, 10 min --> 10 minutes, Line 129, EGC ---> ECG, Line 155, Re-view--> review, Line 302, earl---> early

→ Sorry for the typos. Change made.

  1. Table 1, I guess that n=21 for the controls should be 19? Also the significant values should be highlighted with bold. 

→ Sorry for the confusion. The number of controls for CDP is 21. We changed typos in the manuscript. We highlighted the significant values with bold.

  1. Line 140, Normalized LF/HF ratio were calculated but never explained the min or max scale values what does it mean!

→ The normalized measures are derived or computed indices that are not directly estimated from the raw R-R interval data themselves but are computed as a second step after the initial statistical estimation of the power in the LF and HF bands of the HRV spectrum. Normalization removes most of the very large within-and across-subject variability in the total raw HRV spectral power, which theoretically and empirically tends to follow a long-tailed right-skewed exponential statistical distribution. Moreover, normalization tends to minimize the effect on the values of LF and HF components of the changes in total power. Nevertheless, normalized values should always be quoted with absolute values of LF and HF power in order to describe in total the distribution of power in spectral components. (European Heart Journal 1996)

  1. Line 134, I have concerns about the NN intervals, the authors decided to collect data after 10ms and 20ms. Here I can see that intervals from 0-10ms were omitted and could reveal that patients performed worse/better before it gets to 10ms? could you please elaborate on that?

→ Sorry for the confusion. We did not include pNN10 or pNN20 but pNN50 and HRV triangular index. We changed sentences in the Method section. Ewing and colleagues tested both a fixed threshold of 50 ms and a variable threshold set at 6.25% of the previous NN interval. They recommended the 50 ms fixed threshold because it was “easier and simpler to measure than a percentage threshold”. Subsequently, Bigger and colleagues introduced the pNN50 statistic, defined as NN50 count/total NN count—that is, the percentage of absolute differences in successive NN values > 50 ms. pNN50 has proved to be a useful HRV measure, providing diagnostic and prognostic information in a wide range of conditions. (Heart. 2002)

  1. Line 217-224, should be removed! 

→ Sorry for the careless mistake. We removed the sentences.

  1. The discussion has to be improved and supported with references; 

Line 248, efferent motor copies need to be elaborated!

→ We added sentences in the Discussion section for a detailed explanation about the efferent motor copies according to the reviewer’s comment.

Line 249, need reference! 

→ We added a reference.

  1. I would suggest the authors to have a conclusion section to summarize their results in order to highlight the main findings in the study! 

→ We revised the conclusion section according to the reviewer’s comment.

  1. It would be nice to have figures illustrating the conditions to make it clear for the readers especially with CDP!

→ We added figure 1 illustrating the conditions of CDP according to the reviewer’s recommendation.

Reviewer 2 Report

The aim of this study was to investigate the correlation of PI and autonomic deficits in early PD. Authors recluted 17 patients with a diagnosis of early PD and 19 normal controls. PI was assessed by computerized dynamic posturography (CDP). Standardized autonomic function test (AFT) and time and frequency domain spectral analysis of heart rate variability (HRV) were performed. Findings showed that postural instability in early PD seems to be associated with parasympathetic autonomic dysfunction.

The paper is in general well written and represents a potentially interesting study. However there are some points that should be addressed to clarify the interpretation of the results. They are listed below not necessarily in order of importance.

  • Please include information about analyses performed in methods section of the abstract.
  • Include the number of normal controls in the abstract.
  • In data analysis section, why did the authors use U and Spearman coefficient? Is all the data non-parametric? Which test did the authors use to assess this fact? It would be necessary to assess normality with Shaphiro Wilk test and use parametric or non-parametric tests based on these results.
  • SCOPA-AUT is a scale to evaluate autonomic symptoms outcomes PD that assesses the patient-reported manifestations across six dimensions of autonomic function: digestive, urinary cardiovascular, pupillary, thermoregulatory, and sexual. Did the authors use this scale?
  • Please include a demographic table with PD and normal controls.
  • Apart from heart rate, blood pressure variability, and systolic and diastolic blood pressure were meausured?
  • Sudomotor function was an important biomarker associated to autonomic dysfunction. The authors have made a great effort to explore autonomic dysfunction, but it would be interesting to use more tests to do so.
  • The groups of PD and especially of HC are rather small.
  • Did the authors examine if age of disease onset or disease progression was of importance?
  • Did they observe any significant dependence of the outcome parameters on age?
  • Cognitive dysfunction is a cardinal non-motor feature of PD, and based on MMSE they excluded some PD patients. Did the authors investigate MCI in this sample?
  • It would be interesting to use significantly correlated autonomic variables to perform stepwise linear regression analyses, setting autonomic parameters as independent predictors.
  • Authors explained “Among 17 patients with early PD, all of the early PD was tremor-dominant (n=15) or akinetic rigidity type (n=2)”. I would like to know if the results are maintained if the 2 patients with rigidity type are excluded.
  • Although the study was performed in OFF state, the role of dopamine in the neural mechanisms is highlighted. Did the authors calculate the Levodopa Equivalent Daily Dose for each patient? What was the formula? Please include the reference. I would recommend this article: Tomlinson, C. L., Stowe, R., Patel, S., Rick, C., Gray, R., & Clarke, C. E. (2010). Systematic review of levodopa dose equivalency reporting in Parkinson's disease. Movement disorders, 25(15), 2649-2653.
  • In table 1 and table 2, please include U values.
  • Have the authors performed correlation analysis in normal controls? Please add this information in the methods section.
  • I would recommend examine the effect size (Cohen d and 95% confidence interval [CI]) based on change score differences between groups to investigate the clinically relevance of the results and explain in the discussion.
  • The discussion section would probably benefit if the authors provided more information about the implications of their findings in PD (instead of merely comparing their results with the results of other studies). The authors should also emphasize what the readers can learn from their study.
  • I would like to see the graphs of the correlations.
  • Many of the correlations are only significant at p<.05 and these correlations would presumably not survive a correction for the multiple correlations performed.     
  • I would recommend reviewing the manuscript as there are some typographical errors.

Author Response

  1. Please include information about analyses performed in methods section of the abstract.

→ According to the reviewer’s comment, we added a sentence about statistical analysis methods in the abstract.

  1. Include the number of normal controls in the abstract.

→ We added the number of normal controls in the abstract. The Control group for CDP and AFT were different groups, so we added information for both controls in the Method and results sections.

  1. In data analysis section, why did the authors use U and Spearman coefficient? Is all the data non-parametric? Which test did the authors use to assess this fact? It would be necessary to assess normality with Shaphiro Wilk test and use parametric or non-parametric tests based on these results.

→ We used statistical methods for nonparametric data in our study because we thought the number of patients and controls in each group was small. However, according to the reviewer’s comment, we analyzed all data with Shaphiro Wilk test to assess normality. We confirmed most of the parameters did not fit the normal distribution.

  1. SCOPA-AUT is a scale to evaluate autonomic symptoms outcomes PD that assesses the patient-reported manifestations across six dimensions of autonomic function: digestive, urinary cardiovascular, pupillary, thermoregulatory, and sexual. Did the authors use this scale?

→ Unfortunately, we could not include questionnaires for evaluating autonomic symptoms for early PD patients. In the future study, we could investigate the relationship between autonomic symptoms with questionnaires and postural instability in a larger number of patients. We added sentences about it in the discussion section.

  1. Please include a demographic table with PD and normal controls.

→ We added demographical information for PD and controls in Table 1 and 2.

  1. Apart from heart rate, blood pressure variability, and systolic and diastolic blood pressure were measured?

→ We performed a standardized autonomic function test, including a tilt-table test in all patients. Early PD patients showed mild autonomic dysfunction in a standardized autonomic function test similar to the previous studies. There was no significant correlation between parameters in a standardized autonomic function test and CDP. We did not include these results in the manuscript to focus on the relationship between heart rate variability and CDP.

  1. Sudomotor function was an important biomarker associated to autonomic dysfunction. The authors have made a great effort to explore autonomic dysfunction, but it would be interesting to use more tests to do so.

→ As mentioned above, we performed a standardized autonomic function test, including QSART in all patients. There was no significant correlation between parameters in a standardized autonomic function test and CDP. In the future study, we could investigate the relationship between various autonomic parameters and CDP data with a larger sample size.

  1. The groups of PD and especially of HC are rather small.

→ The sample size was small because we needed to include only early PD with no accompanying other autonomic, cognitive, orthopedic, or vestibular problems. Further studies with a large number of patients will be required to confirm our findings concerning autonomic dysfunction in early PD. We mentioned this in the Limitation section.

  1. Did the authors examine if age of disease onset or disease progression was of importance? Did they observe any significant dependence of the outcome parameters on age?

→ We found that the outcome parameters showed significant dependence on both age and disease progression. Because PD is a degenerative disease, both motor and non-motor symptoms of patients with PD may be inevitably affected by age and disease progression.

  1. Cognitive dysfunction is a cardinal non-motor feature of PD, and based on MMSE, they excluded some PD patients. Did the authors investigate MCI in this sample?

→ We tried to exclude patients with dementia in early PD because we needed to exclude Lewy body dementia or progressive supranuclear palsy. However, we could not exclude patients with MCI in our study. We added a sentence about this in the Limitation section.

  1. It would be interesting to use significantly correlated autonomic variables to perform stepwise linear regression analyses, setting autonomic parameters as independent predictors.

→ Autonomic parameters in HRV are closely related to each other—for example, The RMSSD, pNN50, and HF, which are well-known markers reflecting parasympathetic activity. Therefore, any autonomic parameter could not be independent of other HRV parameters.

  1. Authors explained “Among 17 patients with early PD, all of the early PD was tremor-dominant (n=15) or akinetic rigidity type (n=2)”. I would like to know if the results are maintained if the 2 patients with rigidity type are excluded.

→ After excluding the two patients of rigidity type, the correlation between HRV and CDP showed still positive, but p-value was changed. In the previous study, rigidity type PD showed more autonomic symptoms than tremor type PD (Mov Disord. 2006). We could guess the rigidity type PD has more correlation with autonomic dysfunction and the relationship between HRV and CDP were changed after excluding rigidity type.

  1. Although the study was performed in OFF state, the role of dopamine in the neural mechanisms is highlighted. Did the authors calculate the Levodopa Equivalent Daily Dose for each patient? What was the formula? Please include the reference. I would recommend this article: Tomlinson, C. L., Stowe, R., Patel, S., Rick, C., Gray, R., & Clarke, C. E. (2010). Systematic review of levodopa dose equivalency reporting in Parkinson's disease. Movement disorders, 25(15), 2649-2653.

→  Levodopa equivalent dose (LED) is calculated for all PD patients according to the same method reviewer recommended. The mean levodopa equivalent dose was 412 ± 232mg. We have already described this in the results section.

  1. In table 1 and table 2, please include U values.

→ We added U values in tables.

  1. Have the authors performed correlation analysis in normal controls? Please add this information in the methods section.

→ Unfortunately, normal controls for CDP and AFT were from different groups. So, we could not perform correlation analysis in normal controls.

  1. I would recommend examine the effect size (Cohen d and 95% confidence interval [CI]) based on change score differences between groups to investigate the clinically relevance of the results and explain in the discussion.

→ According to the reviewer’s comment, we calculated 95% confidence interval and add them in table 1 and 2.

  1. The discussion section would probably benefit if the authors provided more information about the implications of their findings in PD (instead of merely comparing their results with the results of other studies). The authors should also emphasize what the readers can learn from their study.

→ According to the reviewer’s comment, we added sentences about the clinical implications of our results in the Discussion section.

  1. I would like to see the graphs of the correlations.

→ We added figure 2 for the correlation graphs.

  1. Many of the correlations are only significant at p<.05 and these correlations would presumably not survive a correction for the multiple correlations performed.

→ As mentioned above, autonomic parameters in HRV are closely related to each other. Composite SOT also is connected to other SOT by its definition, which reflects the overall coordination ability of the visual, vestibular, and somatosensory systems. Therefore, the results from multiple regression analysis cannot but showed no significance.

  1. I would recommend reviewing the manuscript as there are some typographical errors.

→ Sorry for typos. We thoroughly reviewed the entire manuscript and revised typos.
